# Dietary Theabrownin Supplementation Improves Production Performance and Egg Quality by Promoting Intestinal Health and Antioxidant Capacity in Laying Hens

**DOI:** 10.3390/ani12202856

**Published:** 2022-10-20

**Authors:** Tao Zhang, Shiping Bai, Xuemei Ding, Qiufeng Zeng, Keying Zhang, Li Lv, Jian Li, Huanwei Peng, Yue Xuan, Jianping Wang

**Affiliations:** Key Laboratory of Animal Disease-Resistance Nutrition, Ministry of Education, Ministry of Agriculture and Rural Affairs, Key Laboratory of Sichuan Province, Animal Nutrition Institute, Sichuan Agricultural University, Chengdu 611130, China

**Keywords:** laying hens, Pu-erh tea theabrownin, laying performance, intestinal health, antioxidant capacity

## Abstract

**Simple Summary:**

Eggs are one of the main sources of protein for human beings, and the health of laying hens is closely related to egg production. It has been widely accepted that theabrownin has antioxidative functions, regulating microbiota community and lipid metabolism, but studies on layers are scarce. In this study, we aimed to investigate the effects of dietary theabrownin supplementation on production performance, egg quality, intestinal health, and antioxidant capacity in laying hens. Our results revealed that the addition of 100 mg/kg theabrownin in the diet could improve performance and egg quality, and extend the shelf life of eggs. On the other hand, the addition of theabrownin increased the antioxidant capacity of the ovary and the magnum and improved intestinal health. These results revealed that dietary theabrownin supplementation was beneficial to improve production performance and egg quality.

**Abstract:**

Theabrownin, an activated and ample pigment in Pu-erh tea, is known to exert antiobesity and antihyperlipidemic effects in humans, mice, and rats. In this study, we aimed to explore the effects of theabrownin (TB) dietary supplementation on production performance, egg quality, intestinal health, and antioxidant capacities in laying hens. In total, 160 Lohmann laying hens (25 weeks old) were randomly split into four groups (each group 40 hens), namely the CONT (control, basal diet + 0 mg/kg TB), TB1 (basal diet + 100 mg/kg TB), TB2 (basal diet + 200 mg/kg TB), and TB4 (basal diet + 400 mg/kg TB) groups, respectively. These were supplemented with TB for 12 weeks. The results showed that the TB1 group exhibited a significantly higher laying rate during 9 to 12 weeks and higher egg weight and feed conversion efficiency (lower FCR) during 5 to 8 weeks and in the overall period (1 to 12 weeks) than the CONT group (*p* < 0.05). Compared with the CONT group, the eggs from the TB4 group had higher albumen height and Haugh unit than those from the other treatment groups after the 8th and 12th week; notably, the same was also observed in the TB1 and TB2 groups but only after the 12th week (*p* < 0.05). The albumen quality (albumen height and Haugh unit) after 3 weeks of storage was significantly higher in the TB1, TB2 and TB4 groups than in the CONT group (*p* < 0.05). Furthermore, TB supplementation lowered the serum levels of total cholesterol and total triglyceride (*p* < 0.05). Expression analysis revealed that TB2 and TB4 groups had reduced expression of tumor necrosis factor-α (*p* < 0.05), while TB1, TB2, and TB4 had significantly decreased expression of interleukin-1β and IL-6 (*p* < 0.05). Conversely, zonula occludens-1, claudin-1, and mucin-2 were upregulated in the TB2 and TB4 groups (*p* < 0.05). Meanwhile, dietary TB supplementation ameliorated the antioxidant status of the ovary and the magnum, showing a significant reduction in malondialdehyde and 8-hydroxydeoxyguanosine levels in the magnum, the upregulation of glutathione in the ovary, and superoxide dismutase and catalase in the magnum (*p* < 0.05). Overall, dietary supplementation with TB (>100 mg/kg) improved production performance and egg storage quality by improving the intestinal health and antioxidant capacities of the reproductive system in laying hens.

## 1. Introduction

Eggs, with high nutritional values, are a popular and widely consumed food. Additionally, they possess antibacterial, antifungal, antiviral, anti-inflammatory, antihypertensive, and antioxidant properties [1]. The quality of eggshells and egg internal material (albumen and yolk quality) are critical parameters for consumer acceptance. In poultry farms, laying hens experience great oxidative stress due to demanding production rates, which makes them vulnerable to external factors (such as heat, crowding, handling, and harmful diet substance, including heavy metals, mycotoxins, oxidized lipids, etc.) decreasing egg quality [2,3,4]. Notably, epigallocatechin-3-gallate was shown to enhance the antioxidant capacity of ovaries in laying hens [5,6]. Therefore, establishing a nutritional strategy that can enhance the quality and shelf life of eggs can greatly help the poultry industry.

Tea (*Camellia sinensis*, Theaceae) is a popular beverage worldwide. The three major types of tea, green, oolong, and black tea contain different levels of fermented tea leaves [7]. Theabrownin (TB) is a water-soluble, macromolecular tea pigment produced by microbial bioconversion of phenolics, polysaccharides, proteins, lipids, and caffeine compounds involving polyphenol oxidase and peroxidase enzymes [8]. TB, a key active substance of fermented tea, exhibits various biological properties providing health benefits against atherosclerosis, diabetes, and allergic reactions [9,10,11]. As mentioned in previous studies, theaflavins (TFs) and thearubigins (TRs), the two main bioactive compounds in tea, are oxidized to theabrownin (TB) during the fermentation process [12]. It is also known that theabrownin is a typical component of Pu-erh tea, so it may be its bioactive substance for lowering cholesterol and blood lipid levels [13,14]. Previous studies revealed that the bioactive compounds in Pu-erh tea are theabrownin and gallic acid during fermentation. Tea extracts and their components such as TB have strong antioxidant properties against cellular reactive oxygen species (ROS) and nitric oxide (NO) [15]. Nevertheless, there are few studies on the effects of TB on the performance, egg quality, and health of laying hens. Therefore, the purpose of this study is to evaluate the influence of TB on the performance, egg quality, intestinal health, and antioxidant capacities of laying hens.

## 2. Materials and Methods

### 2.1. Extraction of Theabrownin

The theabrownin compounds were obtained from Yunnan Tangren Biotechnology Co., Ltd. (Honghe, China), with a purity of 83%. TB extraction was performed following the details indicated in [12].

Briefly, Pu-erh tea powder (3500 g) was suspended into a tenfold volume of absolute ethyl alcohol, mixed for twelve hours, and then vacuum-strained. The residuum was leached with a tenfold volume of boiled distilled water, retained at 83 °C for 20 min with unremitting stirring, and then vacuum-strained. Repeating the same extraction method three times, the extracts were combined, and then the volume was decreased to one-fifth through vacuum evaporation. A series of liquid–liquid extraction processes were adopted to extract the concentrated solution that was obtained before, involving the same volume of extraction with chloroform, ethyl acetate, and n-butanol, two, three, and four times, respectively. The complete volume of the water for layers was evaporated to one-quarter, and then absolute ethyl alcohol was added to a final proportion of 83.1% to immerse the TB coarse extracts.

### 2.2. Birds, Diets, and Management

All the experimental protocols were approved by the Animal Care and Use Committee of the Sichuan Agricultural University (Chengdu, China). In total, 160 Lohman laying hens (25 weeks old) were randomly allocated to 4 treatment groups. For each treatment (n = 40), there were 10 replicates of 4 birds each that were raised in a single cage (45 × 60 cm^2^). Each treatment group received a different level of TB supplementation (TB was mixed with the diet), namely the CONT (control, basal diet + 0 mg/kg TB), TB1 (basal diet + 100 mg/kg TB), TB2 (basal diet + 200 mg/kg TB), and TB4 (basal diet + 400 mg/kg TB) groups, respectively. The entire experimental time was 12 weeks. The layers were fed an entire feeding mixture in a mashed form (Table 1) and were maintained under an environmental controlled room (45–60% relative humidity, 22 ± 2 °C temperature, 16/8 h light/dark cycle). Feed and water were furnished ad libitum.

### 2.3. Sample Collection and Measurements

The egg quantity, the total egg weight, and the number of broken eggs, dirty eggs, and misshaped eggs were recorded every day for each replicate; feed intake was measured weekly. The ratio of total feed intake (g) to total egg weight (g) was named the feed conversion ratio (FCR). Egg production was estimated as an average daily production. At 4, 8, and 12 weeks, 30 eggs (3 eggs/replicate from 10 replicates) from each treatment group were selected for egg quality trait analysis. Additionally, at the end of 12 weeks, 40 eggs (4 eggs/replicate from 10 replicates) from each group were stochastically gathered to evaluate egg quality during storage at 10 °C for 14 and 21 days. At the end of 12 weeks, all 40 hens (one layer per replicate) in the respective groups were individually weighed, and their blood samples were collected from the jugular vein. The samples were centrifuged at 3000× *g* for 15 min, and the obtained serum samples were stored at −20 °C for subsequent analysis. The hens were then sacrificed through CO_2_ suffocation to collect the ovary, magnum, and intestinal mucosa (jejunum and ileum) samples, which were stored at −80 °C for gene expression analysis.

### 2.4. Egg Quality

An egg multitester (EMT-7300, Robotmation Co., Ltd., Tokyo, Japan) was used to evaluate the egg yolk color, albumen height, and Haugh unit. An eggshell force gauge model II (Robotmation Co., Ltd.) was used to assess eggshell strength. An eggshell thickness gauge (Robotmation Co., Ltd.) was used to measure eggshell thickness, including the large end, subsolar region, and small end. The eggshell redness (a*), yellowness (b*), and lightness (L*) were detected via a colorimeter (Minolta CR410 chroma meter, Konica Minolta Sensing Inc., Osaka, Japan).

### 2.5. Serum Characteristic Analysis

The levels of total protein (TP), glucose, total cholesterol (TC), triglyceride (TG), high-density lipoprotein cholesterol (HDL-C), and low-density lipoprotein cholesterol (LDL-C) were measured with an automatic biochemical analyzer (HITACHI-3100, Hitachi Instruments (Shanghai) Co., Ltd., Shanghai, China).

### 2.6. RT-PCR Expression Analysis of Inflammatory Cytokines and Intestinal Barrier-Related Genes

Following the manufacturer’s instructions, the total RNA from jejunal and ileal mucosa (1 layer/replicate, a total of 8 samples from each group) was extracted using a TRIzol reagent, which was purchased from TaKaRa Biotechnology Co., Ltd. (Dalian, China). A PrimeScript RT reagent kit and gDNA Eraser (TaKaRa Biotechnology Co., Ltd., Dalian, China) were used to synthesize the cDNA of samples. The primers listed in Table 2 were purchased from TaKaRa Biotechnology Co., Ltd. (Dalian, China). The SYBR Premix Ex Taq reagents (TaKaRa Biotechnology, Ltd., Dalian, China) and a CFX-96 Real-Time PCR System (Bio-Rad Laboratories, Richmond, CA, USA) were used to perform the real-time quantitative PCR. The variance in RNA input in the respective reactions was corrected via the housekeeping *β-actin* gene. The housekeeping gene was compared with the relative mRNA expression of *ZO-1* (zonula occludens-1), *ZO-2* (zonula occludens-1), *claudin-1*, *claudin-2*, *occludin*, *mucin-2*, *IL-6* (interleukin-6), *IL-1β* (interleukin-1β), and *TNF-α* (tumor necrosis factor-α). Finally, the relative mRNA expressions were calculated using the 2*^−^*^ΔΔCT^ method [16].

### 2.7. Antioxidant Parameters in Ovary and Magnum

The stored ovary and magnum tissues were thawed on ice. The sample (0.1 g) was mixed with saline (0.9 mL) to make 10% homogenate. The respective homogenates were centrifuged at 3000× *g* and 4 °C for 15 min. The collected supernatants were used to estimate the antioxidant capacities, including antioxidant enzyme activities of catalase (CAT), glutathione peroxidase (GSH-Px), superoxide dismutase (SOD), and the levels of glutathione (GSH), malondialdehyde (MDA), and 8-hydroxydeoxyguanosine (8-OHDG), using the quantitative colorimetric assay kits supplied by Nanjing Jiancheng Bioengineering Institute, Nanjing, China.

### 2.8. Statistical Analysis

In this study, the one-way ANOVA in SPSS software (SPSS 13.0 for Windows, SPSS, Inc., Chicago, IL, USA) was used to analyze the data. Each cage was used as the analysis unit and random factor. When a treatment exhibited a significant effect, the diversities between data averages were evaluated by Duncan’s multiple range analysis. The diversities were considered prominent at *p* ≤ 0.05.

## 3. Results

### 3.1. Production Performance

We found that the TB1 group showed a significantly higher laying rate during 9 to 12 weeks with improved egg weight and feed efficiency (*p* < 0.05). Lower FCR was observed only for 1–12 weeks (*p* < 0.05) (Table 3).

### 3.2. Fresh and Storage Egg Quality

As shown in Table 4 and Table 5, at 8 and 12 weeks, the eggs from the TB4 group exhibited higher albumen height and Haugh unit than those from the other groups (*p* < 0.05). Notably, compared with the control group, at 12 weeks, the albumen height and Haugh unit in TB1 and TB2 groups were also improved (*p* < 0.05). The albumen quality levels (albumen height and Haugh unit) of TB1, TB2, and TB4 treatment groups were remarkably higher than that of the control group after 3 weeks of storage (*p* < 0.05).

### 3.3. Serum Biochemistry Parameters

As revealed in Table 6, compared with the control group, TB supplementation in laying hens lowered their serum TC, LDL-C, and total TG levels (*p* < 0.05). At the same time, the TP was increased in TB4, compared with CONT (*p* < 0.05).

### 3.4. Level of Intestinal Barrier-Related Genes and Inflammatory Cytokines

Compared with the CONT group, the expression of TNF-α in TB2 and TB4 groups significantly decreased (*p* < 0.05), while the mRNA expressions of proinflammatory cytokines (IL-6 and IL-1β) were found to be significantly reduced in the TB1, TB2, and TB4 groups (Figure 1; *p* < 0.05). Conversely, TB1 and TB2 groups showed increased mRNA expressions of barrier-associated proteins (claudin-1, ZO-1, and mucin-2) (Figure 2; *p* < 0.05).

### 3.5. Antioxidant Capacities of Ovary and Magnum Tissues

The effect of TB on the antioxidant capacities of the ovary and magnum tissues is revealed in Table 7. Compared with the CONT group, the addition of TB in the basal diet reduced the levels of MDA and 8-OHDG, while increasing the activities of SOD and CAT in the magnum (*p* < 0.01). Additionally, the GSH level was prominently increased in the ovary tissue (*p* < 0.05). However, apart from these, there was no significant distinction in other determined antioxidant parameters as an effect of TB supplementation (*p* > 0.05).

## 4. Discussion

TB, a key active substance of fermented tea, possesses important functions such as antioxidative, anti-inflammatory, blood-glucose-lowering, and fat-metabolism-regulatory effects. In this study, TB (extracted from Pu-erh tea) dietary supplementation at 100 mg/kg markedly increased the laying rate, egg weight, and feed conversion efficiency in laying hens. Thus far, there are no studies in the literature about the effect of TB on the productive performance of laying hens. However, dietary supplementation with green tea and its extract in laying hens was shown to increase feed efficiency and egg-laying rate [2,5,17]. Similarly, epigallocatechin-3-gallate intervention at 200–400 mg/kg increased egg production in heat-stressed quails [18]. A study showed that supplementation with 200 mg/kg of tea polyphenols improved hen performance during the late laying period [19]. The ovary is a vital organ for laying hens and determines egg production. In the present study, TB supplementation ameliorated ovarian function by upregulating the GSH level and thereby increasing egg production. The beneficial effect of TB on the egg weight and laying rate may be due to the improved ovarian function of laying hens.

We also observed that 100–400 mg/kg TB increased the albumen quality (higher albumen height and Haugh unit) in fresh and stored eggs for up to 21 days. Likewise, other studies have also shown that green tea powder increased the albumen height and Haugh unit in laying hens [2,4,5,20]. A previous study showed that the addition of 200 mg/kg of tea polyphenols in the diet increased the albumen quality in laying hens [19]. Epigallocatechin-3-gallate supplementation increased the albumen height and Haugh unit in laying hens [5]. Our results showed that TB supplementation decreased MDA and 8-OHDG levels while increasing the SOD, GSH, and CAT levels, thereby improving the antioxidant status of the magnum. At the same time, it has been widely known that the albumen is mainly formed in the magnum. Therefore, TB enhanced the antioxidant capacity of the magnum, thus influencing the composition of albumen. The antioxidant capacity of albumen was increased, which improved the quality of the eggs. Taken together, TB supplementation in laying hens can promote egg albumen quality and shelf life.

The nutrient absorption capacity was determined via intestinal morphology, and intestinal morphology is closely related to immune response [21]. It has been suggested that TB supplementation can increase intestinal health by promoting the levels of mucus barrier-associated proteins. This is also the reason for the improvement of laying hens’ performance in this study. Moreover, a foregone study showed that the growth of intestinal probiotics was promoted, and pathogenic bacteria were inhibited via feeding Pu-erh tea extraction, thereby allaying intestinal inflammation [22]. Microbes, which are associated with proinflammatory, can be decreased, and dysbiosis induced by the disease can be counteracted in colitis models via polyphenols [23], thereby promoting the health of the intestine. On the other hand, another study reported that epigallocatechin gallate (EGCG) could attenuate lipopolysaccharide-stimulated acute lung injury in mice and restrained the liberation of inflammatory cytokines *TNF-α*, *IL-1β*, and *IL-6* [24]. Meanwhile, epigallocatechin-3-gallate was found to be available for preventing intestinal disorders or bacterial infections via heightening the immunologic barrier function of the intestine [25,26]. Here, we also found that the relative mRNA expressions of proinflammatory cytokines (*TNF-α*, *IL-1β*, and *IL-6*) in the jejunum were downregulated, while the barrier-associated protein genes (*ZO-1*, *claudin-1*, and *mucin-2*) were upregulated in laying hens in the 100 and 400 TB mg/kg supplementation groups. These results revealed that TB can directly promote intestinal health by enhancing intestinal barrier function and decreasing intestinal inflammation in laying hens. However, the relationship between the ovary and egg production needs to be further investigated.

The capacity of Pu-erh tea with antiobesity and antihyperlipidemic are well-reported, showing a reduction in body weight, serum TC, LDL-C, and TG levels in rats and mice [12,27]. In this study, dietary TB supplementation dramatically decreased the levels of TC, TG, and LDL-C in serum, too. These results indicated that the TB diet may attenuate lipid deposition, thereby promoting the health of lipid metabolism in laying hens. Overall, the addition of TB could improve ovarian function, the antioxidant capacity of the magnum, intestinal health, egg quality, and lipid metabolism. Nevertheless, the relationships between ovarian function, intestinal health, and lipid metabolism are not distinct. Further research is valuable to illustrate the correlation between ovarian function, intestinal health and lipid metabolism, and the underlying mechanism(s).

## 5. Conclusions

This study showed that dietary TB supplementation (100, 200, and 400 mg/kg) in laying hens can improve production performance (egg-laying rate, egg weight, and feed efficiency) and egg quality (egg albumen quality). Mechanistically, it improves intestinal health by upregulating barrier-related proteins, lowering proinflammatory cytokines, and promoting the antioxidant capacities of the ovary and magnum in laying hens. However, the additional level of 100 mg/kg is enough to promote the health and performance of laying hens.

## Figures and Tables

**Figure 1 animals-12-02856-f001:**
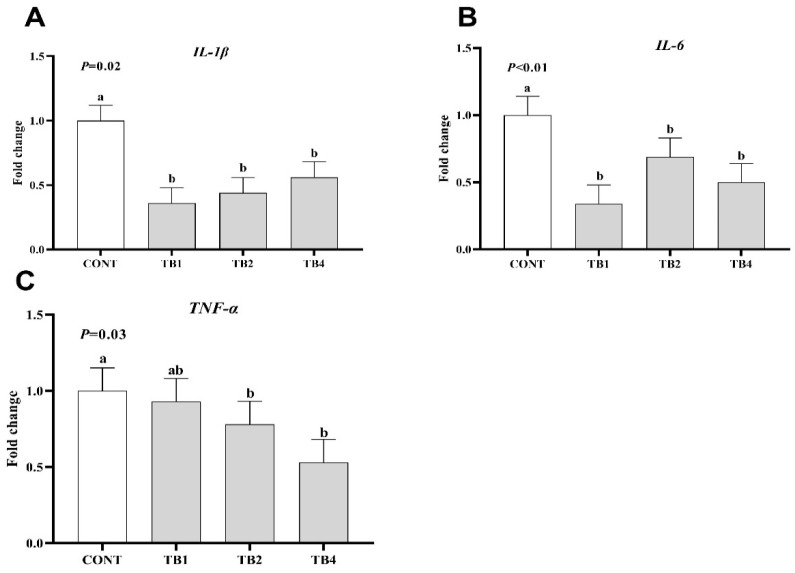
Effects of theabrownin (TB) on inflammatory cytokine mRNA expression in jejunum of laying hens. Data are presented with means and SEM (standard error of means), and bars with diverse alphabets indicate prominently different values (*p <* 0.05). Each mean represents 1 layer/replicate and 10 replicates/treatment. (**A**–**C**) The relative mRNA expression of *IL-1**β*, *IL-6* and *TNF-α* in jejunum respectively. Abbreviations indicate CONT = control group; TB1 = CONT + 100 mg/kg theabrownin; TB2 = CONT + 200 mg/kg theabrownin; TB4 = CONT + 400 mg/kg theabrownin; *IL-1β* = interleukin 1β; *IL-6* = interleukin 6, *TNF-α* = tumor necrosis factor-α.

**Figure 2 animals-12-02856-f002:**
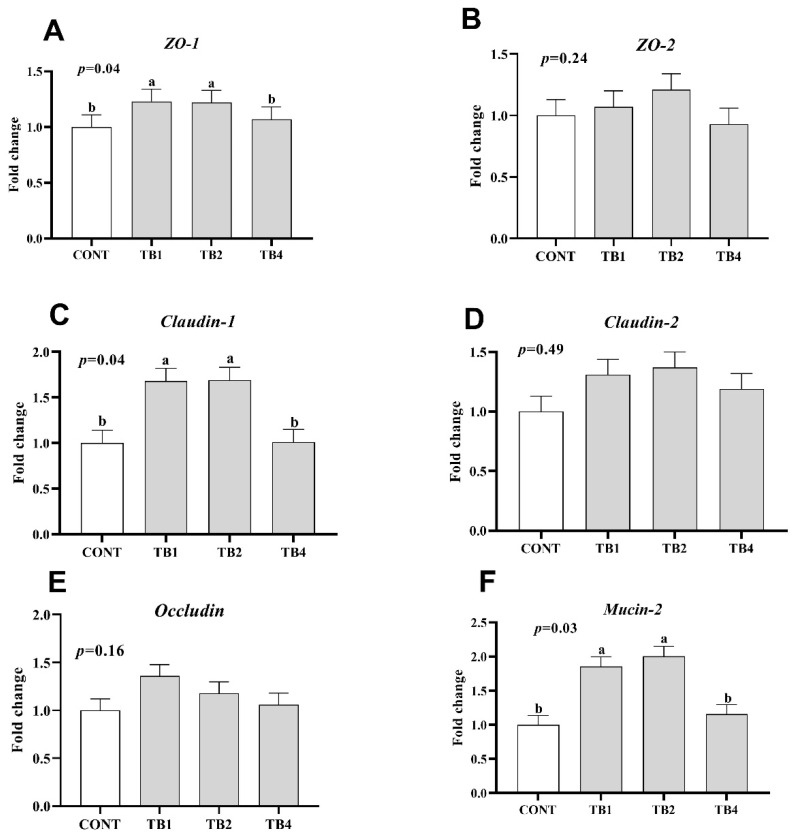
Effect of theabrownin (TB) supplementation on jejunal barrier related gene expression. Data are presented with means and SEM (standard error of means), and bars with diverse alphabets indicate prominently different values (*p* < 0.05). Each mean represents 1 layer/replicate and 10 replicates/treatment. (**A**–**F**) The mRNA expression of *ZO-1*, *ZO-2*, *Claudin-1*, *Claudin-2*, *Occludin* and *Mucin-2* in jejunum respectively. Abbreviations indicate CONT = control group; TB1 = CONT + 100 mg/kg theabrownin; TB2 = CONT + 200 mg/kg theabrownin; TB4 = CONT + 400 mg/kg theabrownin; *ZO-1* = zonula occludens-1; *ZO-2* = zonula occludens-2.

**Table 1 animals-12-02856-t001:** Composition and analysis of basal diet (as-fed basis).

Item, %	Amount
Corn	61.27
Soybean oil Rapeseed meal	2.14 5.00
Soybean meal (CP, 48%)	20.70
Calcium carbonate (granular)	4.47
Calcium carbonate (powder)	4.47
Calcium hydrophosphate	0.59
NaCl	0.21
NaHCO_3_	0.20
L-lysine	0.08
DL-methionine	0.24
Choline chloride	0.10
Vitamin premix ^1^	0.03
Mineral premix ^2^	0.50
Total	100.00
Calculated nutrient content, %	
ME ^3^, kcal/kg	2770
Analyzed nutrient levels, %	
Crude protein	16.55
Calcium	3.53
Total phosphate	0.59
Total lysine	0.80
Total methionine	0.42

^1^ Vitamin premix provides per kg feed vitamin A 10,000 IU; vitamin D_3_ 3000 IU; vitamin E 30 IU; vitamin B_1_ 3 mg; vitamin B_2_ 9.6 mg; vitamin B_6_ 6 mg; vitamin B_12_. 0.3 mg; frolic acid 1.5 mg; calcium D-pantothenate 18 mg; nicotinamide 60 mg; and biotin 1.665 mg. ^2^ Mineral premix provides per kg feed Mn (MnSO_4_) 60 mg; Zn (ZnSO_4_), 80 mg; Cu (CuSO_4_·5H_2_O), 8 mg; Fe (FeSO_4_), 60 mg; I (KI), 0.35 mg; and Se (Na_2_SeO_3_) 0.3 mg. ^3^ Metabolisable energy: was calculated according to NRC 1994.

**Table 2 animals-12-02856-t002:** Primer sequences used to measure gene expression.

Genes	Orientation	Primer Sequences (5′-3′)	Accession Number	Product Size (bp)
*β-actin*	Forward	ATCCGGACCCTCCATTGTC	NM_205518.1	152
	Reverse	AGCCATGCCAATCTCGTCTT		
*Claudin-1*	Forward	GTCTTTGGTGGCGTGATCTT	NM_001013611.2	117
	Reverse	TCTGGTGTTAACGGGGTGTGA		
*Claudin-2*	Forward	CTTTGCTTCATCCCACTGGT	NM_001277622.1	86
	Reverse	TCAAATTTGGTGCTGTCAGG		
*ZO-1*	Forward	GGCAAGTTGAAGATGGTGGT	XM_015278981.2	135
	Reverse	ATGCCAGCGACTGAATTTCT		
*ZO-2*	Forward	AGCAGACCCTGCTCAACATT	NM_204918.1	124
	Reverse	GGGGAGAACGATCTGTTTGA		
*Mucin-2*	Forward	ACCAAGCAGAAAAGCTGGAA	NM_001318434.1	80
	Reverse	AAATGGGCCCTCTGAGTTTT		
*Occludin*	Forward	GCTGAGATGGACAGCATCAA	NM_205128.1	97
	Reverse	TGCCACATCCTGGTATTGAG		
*TNF-α*	Forward	AGATGGGAAGGGAATGAACC	XM_040647309.1	139
	Reverse	GGAAGGGCAACTCATCTGAA		
*IL-1β*	Forward	ACTGGGCATCAAGGGCTA	NM_205518.9	117
	Reverse	GGTAGAAGATGAAGCGGGTC		
*IL-6*	Forward	AGGACGAGATGTGCAAGAAGT	NM_205518.10	98
	Reverse	TTGGGCAGGTTGAGGTTGTT		

Abbreviation indicated: *ZO-1* = zonula occludens-1, *ZO-2* = zonula occludens-2, *IL-1β* = interleukin 1β; *IL-6* = interleukin 6, *TNF-α* = tumor necrosis factor-α.

**Table 3 animals-12-02856-t003:** Effect of dietary theabrownin supplementation on performance in laying hens ^1^.

Group	Laying Rate, %	Egg Weight, g	ADFI, g	FCR	Dirty Egg, %	Broken Egg, %	Misshapen Egg, %
1–4 wk							
CONT	99.01	56.77	113.24	2.04	2.43	0.46	1.08
TB1	99.50	57.68	113.63	1.99	1.79	0.41	1.6
TB2	99.11	56.91	115.64	2.05	2.15	0.63	1.35
TB4	99.11	57.35	114.50	2.03	2.24	1.35	1.18
SEM	0.43	1.31	1.68	0.03	0.86	0.51	0.64
*p*-Value	0.691	0.255	0.496	0.062	0.903	0.231	0.870
5–8 wk							
CONT	99.55	59.28 ^b^	113.23	1.92	3.40	0.27	0.72
TB1	99.41	60.54 ^a^	113.73	1.91	1.69	0.79	1.12
TB2	99.64	59.47 ^ab^	114.51	1.93	2.41	0.27	0.8
TB4	99.02	60.29 ^ab^	114.22	1.91	1.81	0.63	0.81
SEM	0.35	0.55	1.18	0.02	1.16	0.43	0.55
*p*-Value	0.290	0.028	0.721	0.454	0.455	0.531	0.905
9–12 wk							
CONT	96.54 ^b^	60.52	113.49	1.93	2.56	0.28	0.40
TB1	98.81 ^a^	61.86	115.66	1.91	2.63	0.20	0.01
TB2	97.12 ^ab^	60.74	113.89	1.95	2.45	0.47	0.45
TB4	97.86 ^ab^	61.79	114.31	1.90	1.55	0.37	0.38
SEM	0.97	0.67	1.23	0.03	1.49	0.29	0.27
*p*-Value	0.028	0.109	0.328	0.261	0.874	0.802	0.381
1–12 wk							
CONT	98.45	58.85 ^b^	113.32	1.96 ^ab^	2.79	0.34	0.74
TB1	99.24	60.41 ^a^	114.34	1.92 ^c^	2.03	0.46	0.91
TB2	98.64	59.68 ^ab^	114.68	1.99 ^a^	2.34	0.46	0.87
TB4	98.66	59.81 ^ab^	114.34	1.94 ^bc^	1.87	0.79	0.79
SEM	0.49	0.64	1.24	0.02	0.77	0.28	0.35
*p*-Value	0.080	0.017	0.712	0.010	0.649	0.453	0.964

^a–c^ Within a column, means with diverse superscripts differ prominently (*p* ≤ 0.05). ^1^ Each mean represents one group that has 10 replicates and one replicate that has 4 layers. Abbreviations represent CONT = basal diet, TB1 = 100 mg/kg theabrownin, TB2 = 200 mg/kg theabrownin, TB4 = 400 mg/kg theabrownin, FCR = feed conversion ratio, ADFI = average daily feed intake, SEM = standard error of means.

**Table 4 animals-12-02856-t004:** Effect of dietary theabrownin supplementation on egg quality in laying hens ^1^.

Group	Eggshell Strength, kg/cm^3^	Eggshell Thickness, mm	Relative Eggshell Weight, %	Albumen Height, mm	Yolk Color	Haugh Unit	Relative Albumen Weight, %	Relative Yolk Weight, %	Eggshell Color		
									L*	a*	b*
4 wk											
CONT	4.99	3.11	10.93	8.26	5.76	91.00	63.64	25.26	75.09	7.89	22.44
TB1	4.75	3.01	10.71	8.29	5.82	91.15	64.09	25.29	76.03	7.78	21.97
TB2	5.03	3.00	10.91	8.27	5.87	91.01	63.51	25.73	75.99	7.68	22.17
TB4	4.75	2.97	10.65	8.61	5.63	92.71	64.43	25.10	75.79	7.67	22.11
SEM	0.18	0.07	0.14	0.24	0.24	1.11	0.44	0.39	0.86	0.45	0.74
*p*-value	0.284	0.244	0.156	0.136	0.786	0.576	0.161	0.414	0.677	0.951	0.443
8 wk											
CONT	4.65	0.33	10.79	6.78 ^b^	7.22	82.92 ^b^	62.54	26.64	72.48	8.36	22.46
TB1	4.71	0.32	10.72	7.16 ^b^	7.14	84.90 ^b^	63.17	26.10	74.13	7.84	21.56
TB2	4.35	0.34	10.86	7.18 ^b^	7.28	84.05 ^b^	63.10	26.02	74.29	8.34	21.65
TB4	4.32	0.34	10.56	8.04 ^a^	6.91	89.38 ^a^	63.47	25.94	72.77	8.89	23.28
SEM	0.28	0.01	0.13	0.25	0.22	1.63	0.44	0.41	1.48	0.66	0.79
*p*-value	0.066	0.284	0.174	0.022	0.172	0.002	0.259	0.346	0.415	0.323	0.177
12 wk											
CONT	5.08	0.40	11.22	7.47 ^c^	7.03	86.39 ^c^	61.94	26.82	75.40	7.82	20.95
TB1	4.58	0.40	11.40	7.83 ^b^	7.06	88.41 ^bc^	62.01	26.58	76.41	7.21	20.42
TB2	5.04	0.40	11.89	7.87 ^b^	6.96	89.85 ^ab^	61.23	26.55	76.82	7.11	20.07
TB4	4.97	0.39	11.27	8.12 ^a^	6.77	90.84 ^a^	62.11	26.81	76.26	6.91	20.49
SEM	0.20	0.01	0.31	0.16	0.15	0.99	0.58	0.41	0.89	0.52	0.91
*p*-value	0.222	0.896	0.204	0.031	0.239	0.001	0.428	0.866	0.446	0.345	0.815

^a–c^ Within a column, means with diverse superscripts differ prominently (*p* ≤ 0.05). ^1^ Each mean represents one group that has 10 replicates and one replicate that has 3 eggs. Abbreviations represent CONT = basal diet, TB1 = 100 mg/kg theabrownin, TB2 = 200 mg/kg theabrownin, TB4 = 400 mg/kg theabrownin, L* = eggshell lightness, a* = redness, b* = yellowness, SEM = standard error of means.

**Table 5 animals-12-02856-t005:** Effect of dietary theabrownin supplementation on egg quality after storage for 21 days ^1^.

Group	Eggshell Strength, kg/cm^3^	Eggshell Thickness, mm	Relative Eggshell Weight, %	Albumen Height, mm	Yolk Color	Haugh Unit	Relative Albumen Weight, %	Relative Yolk Weight, %	Eggshell Color		
									L*	a*	b*
14 d after storage											
CONT	5.01	0.38	10.56	5.72	6.77	73.79	61.28	28.62	78.53	9.29	24.51
TB1	5.02	0.38	10.59	5.60	6.72	72.12	60.81	28.99	77.86	9.02	24.93
TB2	5.30	0.39	10.82	5.40	6.71	74.82	60.65	28.71	78.62	8.94	24.53
TB4	5.28	0.39	10.54	5.77	6.54	73.89	61.68	28.62	77.37	9.54	24.92
SEM	0.14	0.01	0.13	0.17	0.21	1.07	0.63	0.43	0.96	0.58	0.85
*p*-value	0.062	0.958	0.163	0.163	0.691	0.126	0.363	0.815	0.528	0.728	0.928
21 d after storage											
CONT	4.97	0.37	10.66	4.07 ^b^	7.01	59.06 ^c^	60.79	28.73	78.94	7.07	23.16
TB1	4.85	0.38	10.66	5.14 ^a^	6.76	70.09 ^a^	60.55	28.93	77.20	7.86	22.53
TB2	5.07	0.39	10.79	4.94 ^a^	6.87	66.92 ^b^	60.74	28.46	79.06	8.00	22.82
TB4	4.81	0.38	10.86	5.43 ^a^	6.60	69.41 ^a^	60.39	28.60	78.48	7.87	23.35
SEM	0.18	0.01	0.17	0.27	0.20	1.18	0.73	0.66	1.21	0.64	1.14
*p*-value	0.487	0.193	0.597	0.001	0.228	0.007	0.945	0.908	0.451	0.462	0.372

^a–c^ Within a column, means with diverse superscripts differ prominently (*p* ≤ 0.05). ^1^ Each mean represents one group that has 10 replicates and one replicate that has 3 eggs. Abbreviations represent CONT = basal diet, TB1 = 100 mg/kg theabrownin, TB2 = 200 mg/kg theabrownin, TB4 = 400 mg/kg theabrownin, L* = eggshell lightness, a* = redness, b* = yellowness, SEM = standard error of means.

**Table 6 animals-12-02856-t006:** Effect of dietary theabrownin supplementation on serum characteristics in laying hens ^1^.

Item	Glucose	HDL-C	LDL-C	TC	TG	TP
CONT	13.24	1.20	1.24 ^a^	5.89 ^a^	15.15 ^a^	39.21 ^b^
TB1	12.78	1.27	0.62 ^b^	4.24 ^b^	9.10 ^b^	39.48 ^b^
TB2	12.97	1.31	0.67 ^b^	4.95 ^b^	8.12 ^b^	41.77 ^ab^
TB4	13.03	1.52	0.94 ^b^	3.83 ^c^	7.46 ^b^	47.48 ^a^
SEM	0.39	0.18	0.14	0.68	0.94	2.81
*p*-Value	0.707	0.338	0.038	0.027	0.002	0.025

^a–c^ Within a column, means with diverse superscripts differ prominently (*p* ≤ 0.05). ^1^ Each mean represents one group that has 10 replicates and one replicate that has 1 layer. Abbreviations represent CONT = basal diet, TB1 = 100 mg/kg theabrownin, TB2 = 200 mg/kg theabrownin, TB4 = 400 mg/kg theabrownin, HDL-C = high-density lipoprotein cholesterol, LDL-C = low-density lipoprotein cholesterol, TC = total cholesterol, TG = triglyceride, TP = total protein, SEM = standard error of means.

**Table 7 animals-12-02856-t007:** Effect of dietary theabrownin supplementation on ovary and magnum antioxidant capacity in laying hens ^1^.

Item	Ovary						Magnum					
	GSH μmol/L Prot	GSH-PX U/mL Prot	SOD U/mL Prot	CAT U/mL Prot	MDA nmol/mL	8-OHDG ng/mL	GSH μmol/L Prot	GSH-PX U/mL Prot	SOD U/mL Prot	CAT U/mL Prot	MDA nmol/mL	8-OHDG ng/mL
CONT	121.25 ^b^	334.17	95.91	7.96	1.48	3.91	82.88 ^b^	260.43	229.34 ^b^	95.02 ^b^	1.40 ^a^	17.43 ^a^
TB1	174.71 ^a^	336.84	108.99	10.57	1.25	5.73	98.38 ^b^	256.61	268.39 ^b^	112.57 ^b^	0.81 ^b^	17.10 ^a^
TB2	181.17 ^a^	290.15	99.01	7.64	1.24	5.16	105.32 ^a^	306.80	274.93 ^b^	124.05 ^b^	0.70 ^b^	15.18 ^a^
TB4	182.92 ^a^	289.93	106.48	9.78	1.18	3.51	115.66 ^a^	315.72	348.80 ^a^	166.44 ^a^	0.57 ^b^	11.47 ^b^
SEM	23.83	26.46	6.95	1.41	0.15	0.96	8.66	36.75	30.81	17.80	0.12	1.15
*p*-Value	0.050	0.150	0.213	0.151	0.265	0.121	0.001	0.265	0.001	<0.001	<0.001	0.001

^a,b^ Within a column, means with diverse superscripts differ prominently (*p* ≤ 0.05). ^1^ Each mean represents one group that has 10 replicates and one replicate that has 1 layer. Abbreviations represent CONT = basal diet, TB1 = 100 mg/kg theabrownin, TB2 = 200 mg/kg theabrownin, TB4 = 400 mg/kg theabrownin, GSH = glutathione, GSH-PX = glutathione peroxidase, SOD = superoxide dismutase, CAT = catalase, MDA = malondialdehyde, 8-OHDG = 8-hydroxydeoxyguanosine, SEM = standard error of means.

## Data Availability

Data are available upon request from the corresponding authors.

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
