# Peer review of "Dietary Theabrownin Supplementation Improves Production Performance and Egg Quality by Promoting Intestinal Health and Antioxidant Capacity in Laying Hens"

_animals, 2022, doi:10.3390/ani12202856_

Round 1

Reviewer 1 Report

I started to read the manuscript entitled “Dietary theabrownin supplementation improves production performance and egg quality by promoting intestinal health and antioxidant capacity in laying hens”submitted to Animals as an original article.

However, without clarification some critical issues regarding the experimental design I cannot proceed with a reliable review of the entire manuscript.

Without the information how TB treatment was applied, there is no possibility to define properly the basal experimental unit in this study (hen? replicate cage?). Therefore, it is not possible to tell if the statistical analyses are performed properly and the obtained result are correct and reliable. Tb was applied fer kg of fed or bw ? How it was applied to hens (TB was in lyophilized form)? What was the selection criteria of applied doses ? Was TB identified, its purity and content were somehow verified (any analytical method was applied)?

Minor comments (only introduction section):

L58-59 This sentence is completely out of nowhere.

L70 some additional explanation is needed. In L62 you categorized teas into green, oolong, and black . What type of tea is Pu-erh ?  How it is obtained (something about fermentation process?) ?

Author Response

Dear Editor and Honored Reviewers,

We are so pleased to hear from you. We have corrected our manuscript according to you and reviewer’s suggestion. This is the list of corrections for our manuscript entitled “Dietary theabrownin supplementation improves production performance and egg quality by promoting intestinal health and antioxidant capacity in laying hens”. For your suggestion we have dealt with the comments of the reviewers as follows: All the places which I changed have already been marked yellow highlighting in my paper, for the purpose of highlight. We deeply appreciated all the contribution you all have made, none of our paper could be better without your help. Great efforts have been paid to correct all the backwards as much as we can. Please do let us know, if there are still some other problems.

Editor and Reviewer Comments:

However, without clarification some critical issues regarding the experimental design I cannot proceed with a reliable review of the entire manuscript.

Author response: Had been revised in 2.2.

Without the information how TB treatment was applied, there is no possibility to define properly the basal experimental unit in this study (hen? replicate cage?). Therefore, it is not possible to tell if the statistical analyses are performed properly and the obtained result are correct and reliable.

Author response: Had been revised in 2.2.

Tb was applied fer kg of fed or bw? How it was applied to hens (TB was in lyophilized form)? What was the selection criteria of applied doses? Was TB identified, its purity and content were somehow verified (any analytical method was applied)?

Author response: Had been revised in 2.2 and 2.1.

L58-59 This sentence is completely out of nowhere.

Author response: Sorry, I can not understand your means.

L70 some additional explanation is needed. In L62 you categorized teas into green, oolong, and black. What type of tea is Pu-erh?  How it is obtained (something about fermentation process?)

Author response: Pu-erh tea is a type of fermented tea, during the fermentation process, the catechins and their gallate derivatives are oxidized to complex phenolic tea pigments including theaflavins, thearubigins and, theabrownin (Huang et al., 2019).

Reviewer 2 Report

Dear Authors,

The manuscript can be published after major revision. The experimental design should be described more, the composition of all experimental diets and the chemical characterization of the theabrownin extract should be presented in the paper. Also, the authors need to discuss their own results more.

There are also some minor aspects to be modified: the citation in the text, detailing the chemical methods used, etc.

Please find the following observations:

Line 74-75 : What are the results of that studies?

Line 81: Please follow the guidelines for citation in the text.

Line 82: Where did you get tea powder from? Please describe the specie used, the method/s of drying and processing operation before you obtained the powder.  

Line 91: How did you do that? Using which reagents and equipment? Please describe carefully.

Lines 94-102: The experimental design is a little poor described. Please mention the environmental conditions, the experimental hall, the equipment. What contained the diet before the experiment begins? Do you perform any vaccination schedule? What was the dimension of a cage?

TABLE 1: The total of the ingredients does not reach 100. Please review the ingredients and their percent of inclusion in the diet. Please add the ingredients for each diet: TB0, TB1, TB2, TB4.

Line 115: All 40 hens/group were sacrificed?

Line 116: Was the technique used for direct slaughter, without stunning?  How many samples did you collect?

Lines 127-130: Please describe the method used.

Line 149: Did you mean saline solution? Please mention the concentration of the saline sollution.

Line 164: The affirmation is incorrect. The egg weight was improved in TB1 compared to CONT and FCR was lower in TB1 compared to CONT and TB2 during 1-12 weeks (not 9-12 wk).

Biochemistry parameters lines 3-4: Table 6 shows that also TP recorded differences. Please mention them.

Discussion:

Lines 51-54: Please discuss more your own results. You have cited an article twice in one sentence. PLease pay attention to the citation rules of the journal.

Lines 68-70: the same as previous comment (lines 51-54).

Author Response

Dear Editor and Honored Reviewers,

We are so pleased to hear from you. We have corrected our manuscript according to you and reviewer’s suggestion. This is the list of corrections for our manuscript entitled “Dietary theabrownin supplementation improves production performance and egg quality by promoting intestinal health and antioxidant capacity in laying hens”. For your suggestion we have dealt with the comments of the reviewers as follows: All the places which I changed have already been marked yellow highlighting in my paper, for the purpose of highlight. We deeply appreciated all the contribution you all have made, none of our paper could be better without your help. Great efforts have been paid to correct all the backwards as much as we can. Please do let us know, if there are still some other problems.

Editor and Reviewer Comments:

The manuscript can be published after major revision. The experimental design should be described more, the composition of all experimental diets and the chemical characterization of the theabrownin extract should be presented in the paper. Also, the authors need to discuss their own results more.

Author response: Had been revised in 2.2 and discussion (line 58-63, 82-84 and 90-95). The chemical characterization and function were introduced in line 62-75.

Line 74-75: What are the results of that studies?

Author response: That study showed that the components of fermented tea, involving catechins, theabrownins and caffeine, had different NO Scavenging Abilities and antioxidant properties.

Line 81: Please follow the guidelines for citation in the text.

Author response: Had been revised in line 81.

Line 82: Where did you get tea powder from? Please describe the specie used, the method/s of drying and processing operation before you obtained the powder.  

Author response: These had been described in 2.1.

Line 91: How did you do that? Using which reagents and equipment? Please describe carefully.

Author response: this section have been revised properly, please see line 82-93.

Lines 94-102: The experimental design is a little poor described. Please mention the environmental conditions, the experimental hall, the equipment. What contained the diet before the experiment begins? Do you perform any vaccination schedule? What was the dimension of a cage?

Author response: Had been revised in lines 96-104.

TABLE 1: The total of the ingredients does not reach 100. Please review the ingredients and their percent of inclusion in the diet. Please add the ingredients for each diet: TB0, TB1, TB2, TB4.

Author response: Had been revised in Table 1.

Line 115: All 40 hens/group were sacrificed?

Author response: Had been revised in line 115 (one layer per replicate).

Line 116: Was the technique used for direct slaughter, without stunning?  How many samples did you collect?

Author response: We used a syringe to draw blood from the jugular vein of the chicken, collecting a total of 10 ml.

Lines 127-130: Please describe the method used.

Author response: Had been revised in line 130.

Line 164: The affirmation is incorrect. The egg weight was improved in TB1 compared to CONT and FCR was lower in TB1 compared to CONT and TB2 during 1-12 weeks (not 9-12 wk).

Author response: laying rate was improved in TB1 compared to CONT and FCR was lower in TB1 during 1-12 weeks.

Biochemistry parameters lines 3-4: Table 6 shows that also TP recorded differences. Please mention them.

Author response: Had been revised in Biochemistry parameters lines 4-5.

Discussion:

Lines 51-54: Please discuss more your own results. You have cited an article twice in one sentence. PLease pay attention to the citation rules of the journal.

Author response: Had been revised in line 55.

Lines 68-70: the same as previous comment (lines 51-54).

Author response: Had been revised in line 71.

Reviewer 3 Report

The authors have investigated important novel study through Dietary theabrownin supplementation improves production  performance and egg quality by promoting intestinal health and antioxidant capacity in laying hens. The study is interesting and provides novel information on the egg production parameters and capability in laying hens. 

The abbreviated words such as TB cannot be brought into the start of a sentence

authors need to identify which level was the best to give proper conclusion. 

If the authors claim of scarce studies on TB, then on what basis they suggested the given levels for the laying hens??

I think in the introduction, the given materials are not enough to jsutify this study. It is not known what was the age of the birds at which the experiment was started and what is the end date. why egg shell color has been selected as a parameter? why not egg yolk color which is more important. 

Justification of the results in the discussion part is very poor. Less attention has been given to the mechanism through which TB could produce possible positive effects on the given parameters. 

conclusion section is very  weak. It has not been properly mentioned which level produced the best results. Also the authors have not the orthogonal analysis. 

Also in the statistical analysis, duncan multiple test is not a good test now a days. 

Author Response

Dear Editor and Honored Reviewers,

We are so pleased to hear from you. We have corrected our manuscript according to you and reviewer’s suggestion. This is the list of corrections for our manuscript entitled “Dietary theabrownin supplementation improves production performance and egg quality by promoting intestinal health and antioxidant capacity in laying hens”. For your suggestion we have dealt with the comments of the reviewers as follows: All the places which I changed have already been marked yellow highlighting in my paper, for the purpose of highlight. We deeply appreciated all the contribution you all have made, none of our paper could be better without your help. Great efforts have been paid to correct all the backwards as much as we can. Please do let us know, if there are still some other problems.

Editor and Reviewer Comments:

The abbreviated words such as TB cannot be brought into the start of a sentence.

Author response: Had been revised in line 22.

authors need to identify which level was the best to give proper conclusion. 

Author response: Had been revised in line 101-103.

If the authors claim of scarce studies on TB, then on what basis they suggested the given levels for the laying hens??

Author response: We referred to the dosage of other tea extracts.

I think in the introduction, the given materials are not enough to jsutify this study. It is not known what was the age of the birds at which the experiment was started and what is the end date. why egg shell color has been selected as a parameter? why not egg yolk color which is more important. 

Author response: All egg shell color and yolk color had been selected as a parameter, please see the Table 4, 5. The age of birds and the time of experiment please see 2.2.

Justification of the results in the discussion part is very poor. Less attention has been given to the mechanism through which TB could produce possible positive effects on the given parameters. 

Author response: Discussion was revised, please see line 58-63, 82-84 and 90-95.

conclusion section is very weak. It has not been properly mentioned which level produced the best results. Also the authors have not the orthogonal analysis. 

Author response: Had been revised in conclusion.

Also in the statistical analysis, duncan multiple test is not a good test now a days. 

Author response: Thanks for your suggest, we will attention it in the future experiment.

Round 2

Reviewer 1 Report

Some editing for English language is required throughout the manuscript due to too many mistakes.

All p-values should be expressed to 3 digits.

L59-60 there is no logical sequence between this sentence and the preceding and following. Where does this chemical come from? From air ? Please remove.

L127 “Machines irradiate eggshells to get data “ – remove

L136 TRIzol is registered trademark by Life technologies (Thermo Fisher).

L139 “"we can see" - please rephase

L147 “expression of ZO-1 et al “ - please rephase

Table 7 – Please add corresponding units to presented parameters.

The beginning of the last paragraph of the discussion - and what about ADFI and (potential) changes in BW in your study ? Please discuss this aspect.

Author Response

Dear Honored Reviewers,

We are so pleased to hear from you. We have corrected our manuscript according to you and reviewer’s suggestion. This is the list of corrections for our manuscript entitled “Dietary theabrownin supplementation improves production performance and egg quality by promoting intestinal health and antioxidant capacity in laying hens”. For your suggestion we have dealt with the comments of the reviewers as follows: All the places which I changed have already been marked yellow highlighting in my paper, for the purpose of highlight. We deeply appreciated all the contribution you all have made, none of our paper could be better without your help. Great efforts have been paid to correct all the backwards as much as we can. Please do let us know, if there are still some other problems.

Editor and Reviewer Comments:

Some editing for English language is required throughout the manuscript due to too many mistakes.

Author response: The manuscript had been revised carefully.

All p-values should be expressed to 3 digits.

Author response: Had been revised in all tables.

L59-60 there is no logical sequence between this sentence and the preceding and following. Where does this chemical come from? From air? Please remove.

Author response: Epigallocatechin-3-gallate and theabrownin are both tea extracts, and they may be have some similar effects.

L127 “Machines irradiate eggshells to get data “– remove

Author response: Had been removed.

L136 TRIzol is registered trademark by Life technologies (Thermo Fisher)

Author response: The TRIzol were purchased from TaKaRa Biotechnology (Dalian) Co., Ltd. Had been revised in line 137.

L139 “"we can see" - please rephase

Author response: Had been revised in line 139.

L147 “expression of ZO-1 et al “- please rephase

Author response: Had been revised in line 147.

Table 7 – Please add corresponding units to presented parameters.

Author response: Had been revised in Table 7.

The beginning of the last paragraph of the discussion - and what about ADFI and (potential) changes in BW in your study? Please discuss this aspect.

Author response: Usually, after the laying hens reach the peak laying period, the weight of laying hens will not change much. Therefore, we only consider that the weight of laying hens is within a certain range at the beginning of the experiment.

Reviewer 2 Report

Dear Authors,

After your revision, the manuscript was improved, but I have made a few minor observations that should be clarified and that may not have been clear when I first made them.

Line 85, 93: Did you mean ethanol or ethyl alcohol? Use one of them; it is the same. 

Please add the ingredients in Table 1 for each diet: TB0, TB1, TB2, TB4.

100-101: Please replace normal diet to basal diet. Use the same term throughout the article.

Author Response

Dear Editor and Honored Reviewers,

We are so pleased to hear from you. We have corrected our manuscript according to you and reviewer’s suggestion. This is the list of corrections for our manuscript entitled “Dietary theabrownin supplementation improves production performance and egg quality by promoting intestinal health and antioxidant capacity in laying hens”. For your suggestion we have dealt with the comments of the reviewers as follows: All the places which I changed have already been marked yellow highlighting in my paper, for the purpose of highlight. We deeply appreciated all the contribution you all have made, none of our paper could be better without your help. Great efforts have been paid to correct all the backwards as much as we can. Please do let us know, if there are still some other problems.

Editor and Reviewer Comments:

Line 85, 93: Did you mean ethanol or ethyl alcohol? Use one of them; it is the same.

Author response: Had been revised in line 85 and 93.

Please add the ingredients in Table 1 for each diet: TB0, TB1, TB2, TB4

Author response: The Table 1 is the table of basal diet and we add the TB in basal diet with different levels. Meanwhile, the levels of TB was described in abstract and 2.2. So, it is not necessary to add it in Table 1.

100-101: Please replace normal diet to basal diet. Use the same term throughout the article.

Author response: Had been revised in line 27-28 and 100-101.